# Spontaneous Soft Tissue Hematomas in Patients with Coagulation Impairment: Safety and Efficacy of Transarterial Embolization

**Davide Fior** [1], **Stefano Di Provvido** [2], **Davide Leni** [3], **Rocco Corso** [3], **Lorenzo Paolo Moramarco** [1], **Matteo Pileri** [4,5], **Rosario Francesco Grasso** [4,5], **Domiziana Santucci** [4,5] and **Eliodoro Faiella** [4,5,*]

1. Department of Radiology, Sant'Anna Hospital, ASST Lariana, Via Ravona 20, San Fermo della Battaglia, 22042 Como, Italy; davide.fior@asst-lariana.it (D.F.); lorenzo.moramarco@asst-lariana.it (L.P.M.)
2. Department of Radiology, Desio Hospital, ASST Brianza, Via Giuseppe Mazzini 1, Desio, 20832 Monza, Italy; stefano.diprovvido@asst-brianza.it
3. Department of Diagnostic Radiology, San Gerardo Hospital, ASST Monza, Via Gian Battista Pergolesi 33, 20900 Monza, Italy; davide.leni@asst-monza.it (D.L.); rocco.corso@asst-monza.it (R.C.)
4. Unit of Radiology and Interventional Radiology, Fondazione Policlinico Universitario Campus Bio-Medico, Via Alvaro del Portillo, 00128 Rome, Italy; matteo.pileri@unicampus.it (M.P.); r.grasso@policlinicocampus.it (R.F.G.); d.santucci@policlinicocampus.it (D.S.)
5. Research Unit of Radiology, Department of Medicine and Surgery, Università Campus Bio-Medico di Roma, Via Alvaro del Portillo, 00128 Rome, Italy
* Correspondence: e.faiella@policlinicocampus.it; Tel.: +39-06-22541-1669

**Abstract:** The aim of this study is to report the authors' experience of percutaneous transarterial embolization (TAE) in patients with spontaneous soft tissue hematomas (SSTH) and active bleeding with anticoagulation impairment. The study retrospectively identified 78 patients who received a diagnosis of SSTH by CT scan and underwent TAE between 2010 and 2019 in a single trauma center. The patients were stratified using Popov classification into categories: 2A, 2B, 2C, and 3. The patient's 30-day survival after TAE was considered the primary outcome; immediate technical success, the need for additional TAE, and TAE-related complications were considered secondary outcomes. Immediate technical success, complication rate, and risk factors for death were analyzed. Follow-up stopped on day 30 from TAE. 27 patients (35%) fell into category 2A, 8 (10%) into category 2B, 4 (5%) into category 2C, and 39 (50%) into category 3. Immediate technical success was achieved in 77 patients (98.7%). Complications included damage at the arterial puncture site (2 patients, 2.5%) and acute kidney injury (24 patients, 31%). Only 2 patients (2.5%) had been discharged with a new diagnosis of chronic kidney disease. The 30-day overall mortality rate was 19% (15 patients). The mortality rate was higher in hemodynamically unstable patients, in Popov categories 2B, 2C, and 3, and in patients with an initial eGFR < 30 mL/min × 1.73 m$^2$. The study demonstrated a higher mortality risk for categories 2B, 2C, and 3 compared to category 2A. Nonetheless, TAE has proven effective and safe in type 2A patients. Even though it is unclear whether type 2A patients could benefit from conservative treatment rather than TAE, in the authors' opinion, a TAE endovascular approach should be promptly considered for all patients in ACT with active bleeding demonstrated on CT scans.

**Keywords:** spontaneous soft tissue hematoma (SSTH); anticoagulant therapy (ACT); transarterial embolization (TAE); CT angiography

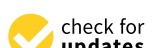



## 1. Introduction

Spontaneous soft tissue hematomas (SSTH) are a potentially severe complication of anticoagulant therapy (ACT) and are associated with other causes of coagulation impairment. Hematomas are defined as "spontaneous" when they are unrelated to evident trauma or surgery in the previous three weeks or to an underlying disease (e.g., infectious

disease, chronic liver diseases, blood clotting disorders, or cancer). The incidence of SSTH is constantly rising as the number of patients undergoing ACT increases every year [1–5].

Risk factors for SSTH include specific comorbidities (e.g., inflammatory conditions, aging, or radiation therapy), but the pathophysiology is still unknown. A diffuse microvascular origin has been postulated due to preexisting diffuse small-vessel arteriosclerosis or heparin-induced microangiopathy. Minor or unrecognized trauma, such as coughing fits, twisting, rapid changes in position, and Valsalva maneuvers, have been reported to be precipitating factors that may have a role in the development of rectus sheath hematoma [6].

When the bleeding is contained to some extent by the surrounding fascial and muscle groups, the resulting hematoma is usually self-limited due to tamponade by the surrounding structures. However, if the bleeding is not promptly controlled, it can lead to hemodynamic instability and even compartment syndrome [7].

Clinical diagnosis is challenging at the onset of signs and symptoms, and it is usually correctly performed in only 40% of patients. Computer tomography (CT) and CT angiography (CTA) are mainly used for diagnosis, allowing a detailed study of the retroperitoneal and extraperitoneal spaces [8–11]. At the moment, there are no studies evaluating CTA's diagnostic efficacy for identifying active bleeding [7].

The correct choice of management is still under debate. Initial conservative measures, such as correction of altered coagulation parameters, fluid resuscitation, and blood transfusion, are usually sufficient and remain the standard of care in hemodynamically stable patients. It has been suggested that more aggressive approaches should be reserved for unresponsive and complicated patients [12]. Endovascular management is favored for unstable patients, whereas surgery may be indicated to evacuate a compressive hematoma. However, ligation of the bleeding vessels at surgery may be challenging in a hemodynamically unstable patient and is no longer considered the standard of care in most situations [13]. For this reason, percutaneous transarterial embolization (TAE) has gained acceptance in the past few years, with an increasing number of published case reports and recent case series [14–20].

In 2017, Popov et al. proposed a classification based on clinical conditions, findings at CT scans, and pharmacological anamnesis to stratify the patients into different classes of risk to evaluate the best choice for treatment [17]. Other studies have suggested management algorithms for SSTH based on clinical experience, but validated algorithms addressing patients with active bleeding for optimal treatment still need to be developed [18,20].

This study aims to report our experience treating patients with SSTH and active bleeding with TAE. We stratified our patients using the Popov classification, analyzed the safety and efficacy of TAE, and evaluated possible risk factors for death. Furthermore, we compared CTA findings with digital subtraction angiography (DSA) to evaluate the accuracy of CTA in identifying the bleeding vessel.

## 2. Materials and Methods

### 2.1. Study Population

We retrospectively identified 78 patients (Figure 1) who were diagnosed with SSTH by CT examination and underwent TAE between January 2010 and October 2019 at a tertiary care center with a 24/7 on-call interventional radiology service. Patients who had surgical interventions or trauma within the three weeks before SSTH were excluded because they were considered to have a potentially traumatic hematoma.

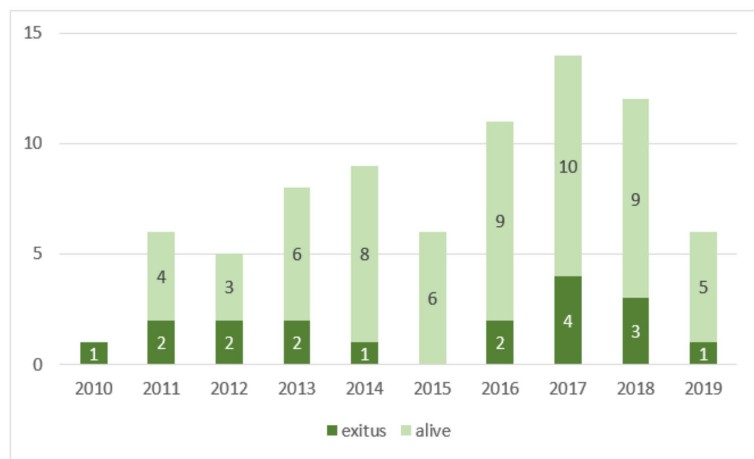

**Figure 1.** Number of patients alive (light green) and dead (exitus, dark green) per year in the study population.

Table 1 summarizes patients' demographic, clinical, and laboratory data.

**Table 1.** Characteristics of patients.

| Patients Data | | |
|---|---|---|
| Patients | | 78 |
| Age | mean ± SD (range) median (Q1–Q3) | 76.3 ± 8.7 (49–93) 78 (71–81) |
| Sex | female:male (%:%) | 51:27 (65:35) |
| Hemodynamic instability | | 23 (29%) |
| Need for transfusion | | 64 (82%) |
| Hemoglobin level (g/dL) | mean ± SD (range) median (Q1–Q3) | 10.0 ± 2.3 (4.6–15.8) 9.7 (8.4–11.5) |
| Platelet count (×1000/uL) | mean ± SD (range) median (Q1–Q3) | 244 ± 112 (36–641) 227 (181–285) |
| INR | mean ± SD (range) median (Q1–Q3) | 1.85 ± 1.71 (0.88–10.64) 1.29 (1.10–1.68) |
| aPTT ratio | mean ± SD (range) median (Q1–Q3) | 1.51 ± 0.99 (0.65–7.38) 1.23 (1.03–1.46) |
| S-creatinine (mg/dL) | mean ± SD (range) median (Q1–Q3) | 1.3 ± 0.9 (0.2–5.4) 1.1 (0.8–1.5) |
| eGFR (mL/min × 1.73$^2$) (CKD-EPI) | mean ± SD (range) median (Q1–Q3) | 57 ± 26 (10–156) 57 (37–78) |

## 2.2. CT Examination

All patients underwent CT examinations before TAE, including an unenhanced CT scan, an arterial phase with bolus-tracking, and a venous phase. Currently, the center is equipped with three 256-row MDCT (iCT Elite; Philips Medical Systems, Eindhoven, The Netherlands) CT scanners. Until 2013, scans were performed on a 16-slice MDCT (Brilliance; Philips Medical Systems, Eindhoven, The Netherlands) CT scanner. CT images were acquired in the transverse plane, with a section thickness of 0.625 mm. Active bleeding, defined as the presence of a contrast blush on arterial phase CT images that increased on venous phase CT images, was documented in all patients (f.e., Figures 2 and 3). A team member (S.D.) retrospectively analyzed the results of the CT examinations: location of the hematoma, fascial rupture, time of the arterial phase scan, and visible bleeding vessels were recorded.

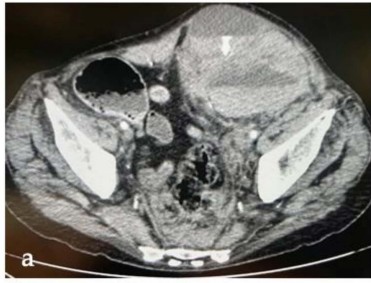 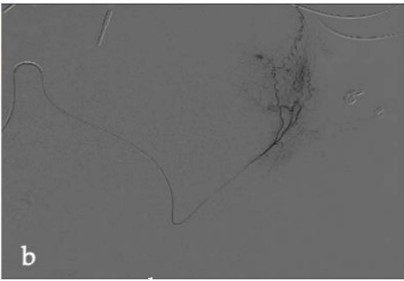 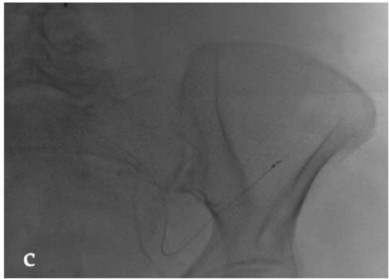

**Figure 2.** LB, a 76-year-old woman under anticoagulant therapy (unfractionated heparin), developed a type 3 intramuscular hematoma in the left psoas muscle. (**a**) CT scan documented an intramuscular hematoma in the left psoas muscle. (**b**) DSA during selective catheterization of the left deep circumflex iliac artery confirmed active bleeding. (**c**) Microcatheter and proximal coiling in the deep circumflex iliac artery. Arteriographic control (not shown) confirmed complete exclusion of the bleeding segment and thus no contrast extravasation.

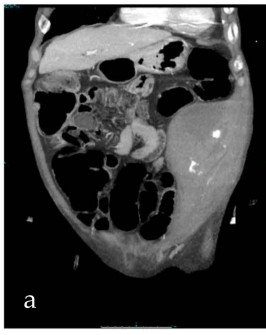 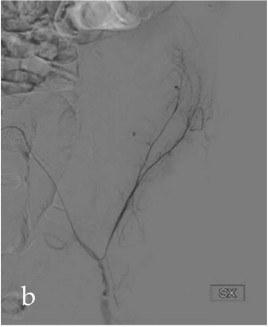 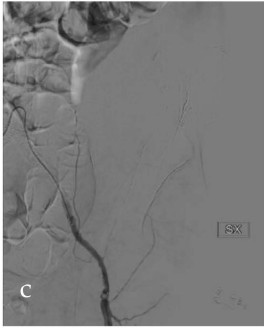 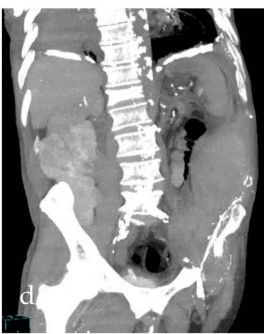

**Figure 3.** RG, an 81-year-old man under therapy with acetylsalicylic acid and rivaroxaban, showed a loss of 2 points of hemoglobin. (**a**) CT scan documented an intramuscular hematoma with active bleeding in the left iliac and transverse muscles. (**b**) DSA confirmed multiple active bleedings from the deep circumflex artery's branches. (**c**) DSA after glue injection, which showed the bleeding interruption. (**d**) CT scan after the procedure demonstrated hematoma reduction and initial resorption. SX = left.

### 2.3. Angiography and TAE Procedure

All patients with active bleeding on computed tomography angiography (CTA) underwent TAE. All procedures were performed by senior interventional radiologists (R.C., D.L., F.V., and D.F.) with extensive arterial embolization experience using a C-ARM angiography system (Allura Xper FD 20; Philips Medical Systems, Eindhoven, The Netherlands).

After local anesthesia of the femoral puncture site with a subcutaneous injection of 5 mL of 2% mepivacaine solution (Pfizer Pharmaceuticals Group, New York, NY, USA), a 5-F vascular sheath introducer (AVANTI®; Cordis, Johnson & Johnson Company, Miami Lakes, FL, USA) was inserted into the common femoral artery.

Selective catheterization of the culprit's vessel was then achieved using different 4- or 5-F catheters (Cobra, Simmons, Vertebral, Berenstein, and Shepherd hook catheters; Cordis, Johnson & Johnson Company, Miami Lakes, FL, USA) and coaxially a 2.7-F microcatheter and a 0.021″ guidewire (Progreat™ Micro Catheter System; Terumo, Tokyo, Japan). If active extravasation of i.v. contrast was confirmed on DSA, the microcatheter tip was advanced as close as possible to the bleeding before embolization.

The embolic agents were chosen by each operator and included fiber platinum microcoils (Vortx™; Boston Scientific, La Garenne Colombes, Cedex, France), detachable microcoils (Concerto™; Medtronic, Minneapolis, MN, USA), absorbable gelatin sponge slurry (Gelfoam®; Pfizer, New York, NY, USA), glue (Glubran® 2, n-butil-2-cyanoacrilate, GEM Srl, Viareggio, Italy, diluition variable), and microfibrillar collagen hemostat slurry (Avitene™; BD-Bard, Franklin Lakes, NJ, USA).

If no active bleeding occurred during DSA, empiric temporary embolization was performed based on DSA imaging suspicion without prior confirmation of the exact origin and location of the bleeding. Reabsorbable embolic agents alone were used empirically when angiography did not confirm active bleeding. The combined use of reabsorbable agents and coils was considered an option for patients with confirmed active bleeding when super-selective catheterization of a target vessel was obtained.

Manual compression or a vascular closure device (Angio-Seal™; Terumo, Tokyo, Japan) achieved hemostasis at the puncture site. Data regarding the time of the first DSA run, the presence of active bleeding at CTA, the type of embolic agent used, and target vessels were recorded.

### 2.4. Popov Classification

As previously mentioned, Popov et al. proposed a classification system to stratify patients into three main categories:

- Type 1: Hemodynamically stable patients without active bleeding on CT;
- Type 2: Patients with active bleeding on CT and without fascial rupture;
- Type 3: Patients with active bleeding extending into the surrounding tissues and fascial rupture.

Type 2 patients were further subcategorized into: type 2A, patients on anticoagulation therapy (ACT) who could stop ACT; type 2B, patients on ACT who had to continue ACT; and type 2C, patients who were not on ACT. The recommended treatment for each category was as follows: conservative treatment (rectification of coagulation parameters, fluid repletion, and blood transfusion) for types 1 and 2A; TAE for types 2B, 2C, and 3 [16].

We applied the same stratification to our population. Of the 78 patients, 27 (35%) were classified as type 2A, 8 (10%) as type 2B, 4 (5%) as type 2C, and 39 (50%) as type 3 (Figure 4).

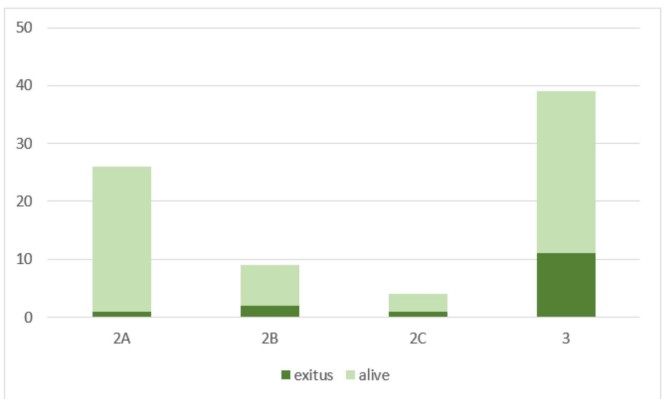

**Figure 4.** Distribution of patients and deaths (dark green) for each category of the Popov classification.

Furthermore, we compared the incidence of active bleeding at DSA examination for the type 2A category and the 2B, 2C, and 3 categories to evaluate if type 2A patients had a higher incidence of self-limited bleeding, defined as the absence of active bleeding signs at DSA examination.

### 2.5. Outcomes

The primary clinical endpoint was the patient's 30-day survival after TAE. Secondary outcomes included immediate technical success, the need for additional TAE, and TAE-related complications such as puncture site damage, non-target embolization, and acute kidney injury (AKI) due to contrast agent-induced nephropathy (CIN). The immediate procedural success was defined as successful catheterization, complete occlusion of the target bleeding vessel, and the absence of active bleeding on the final DSA control. For patients without visible active bleeding, angiographic success was defined as a complete occlusion of the arteries feeding the hematoma territory.

*2.6. Data Analysis, Measurements, and Statistical Analysis*

The statistical analysis was performed using Microsoft Excel® (Microsoft, Redmond, Washington, WA, USA) and IBM SPSS® Statistics 20.0 (IBM, Armonk, New York, NY, USA). Continuous variables were expressed as mean, standard deviation, range, or median, quartile 1, and quartile 3. Qualitative variables were expressed as raw numbers, proportions, and percentages. Categorical variables were compared using the Chi-square test and odds ratio. Overall survival was estimated using the Kaplan-Meier method, calculated from the time of TAE until death due to any cause within 30 days following the procedure. Follow-up information was available for all patients in our study until death or until it was censored after 30 days.

## 3. Results

Our study included a population of 78 patients with a mean age ± standard deviation of 76.3 ± 8.7 y (range, 49–93 y; median age, 78 y; Q1–Q3, 71–81 y), consisting of 51 women (77.2 ± 9.4 y; range, 49–93 y; median age, 79 y; Q1–Q3, 74–83 y) and 27 men (74.7 ± 7.0 y; range, 57–88 y; median age, 75 y; Q1–Q3, 72–79 y) (Figure 5).

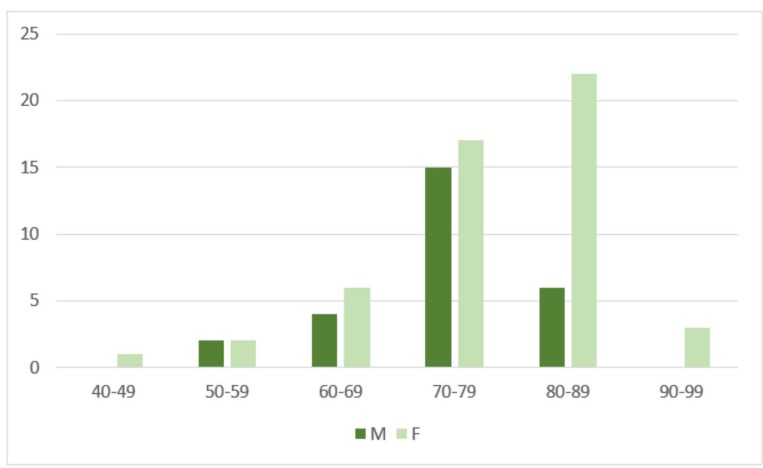

**Figure 5.** Distribution of patients by age and sex (light green for females, dark green for males).

Hematomas were located as follows: 41 (53%) in the rectus sheath, 16 (21%) in the lower limb, 11 (14%) in the iliopsoas muscles, and 11 (14%) in other locations. Among the other locations, 5 patients had hematomas in the lateral abdominal wall, 5 in the upper limb, and 1 in the trapezius muscle. One patient had hematomas in multiple locations (rectus sheath and lower limb).

At admission, 62 patients (79%) were receiving anticoagulant therapy (37 were on heparin, 19 were on apixaban, and 6 were on warfarin), 1 was receiving antiplatelet therapy alone (cardioaspirin), 10 (13%) were receiving both anticoagulant and antiplatelet therapy (heparin and cardioaspirin), and 5 (6%) were receiving no anticoagulant or antiplatelet therapy. Among the 5 patients with no antiplatelet or anticoagulant therapy, 3 had hemophilia A, one had a low platelet count, and one had a coagulopathy of unknown origin.

23 patients (29%) experienced hemodynamic instability that required anesthesiologic support and volume resuscitation. Out of the 78 patients with active bleeding at CTA, only 62 (79%) showed active bleeding at DSA. The median delay between CTA and DSA was 133 min, with a median value of 121 min (Q1–Q3, 89–204 min, 62 patients) in patients with active bleeding at CTA and a median value of 214 min (Q1–Q3, 135–270 min, 16 patients) among patients without it. However, the difference between the two groups was not statistically significant.

We did not observe any statistically significant difference in the incidence of self-limited bleeding at angiography for the type 2A group (14%, 4 patients) compared to the type 2B-2C-3 group (23%, 12 patients). Additionally, the comparison of median de-

lay between CTA and DSA was not statistically significant between the type 2A group (148 min) and the type 2B-2C-3 group (127 min). No statistically significant difference between embolizing agents was found.

Immediate technical success was achieved for all but one patient (98.7%), who died some hours after the procedure.

31 patients underwent a second CTA, and 15 had a second DSA. 13 patients required a second TAE, and the immediate procedural success rate for patients needing a second TAE was 100%.

After 30 days, the mortality rate was 19% (15 patients). During the follow-up, at clinical examination, there was no evidence of nerve or muscle ischemia or infection within the target or nontarget territories related to TAE. Complications at the arterial puncture site occurred twice (2.5%) at the first angiography, requiring stenting in one case. Acute kidney injury (AKI) occurred in 24 patients (31%), 24–72 h after contrast media administration. Only 2 patients (2.5%) had been discharged with a new chronic kidney disease (CKD) diagnosis. The mean contrast amount for all patients was $140 \pm 40$ (mL) 50% dilute with saline solution ($280 \pm 80$ mL of injected solution), with no statistically significant difference between the groups. The mortality rate was higher in hemodynamically unstable patients (9/23, 39%) than in stable patients (6/55, 11%) (OR 5.3, 95% CI 1.6–17.3, $p < 0.01$). The mortality was higher in types 2B, 2C, and 3 (14/52, 27%) than in type 2A (1/26, 4%) (OR 9.2, 95% CI 1.1–74.5, $p < 0.05$) (Figure 6). Moreover, it was higher in patients with initial eGFR < 30 mL/min $\times$ 1.73 m$^2$ (6/12, 50%) compared with other patients (12/66, 18%) (OR 4.5, 95% CI 1.2–16.4, $p < 0.05$). Low eGFR or hemodynamic instability at diagnosis were significant prognostic factors for our patients (OR 4.5, 95% CI 1.2–16.4, $p < 0.05$) (OR 5.3, 95% CI 1.6–17.3, $p < 0.01$). The mortality rate also appeared higher in patients treated with reabsorbable embolic agents alone than in patients treated with both reabsorbable agents and coils; however, these results were not statistically significant.

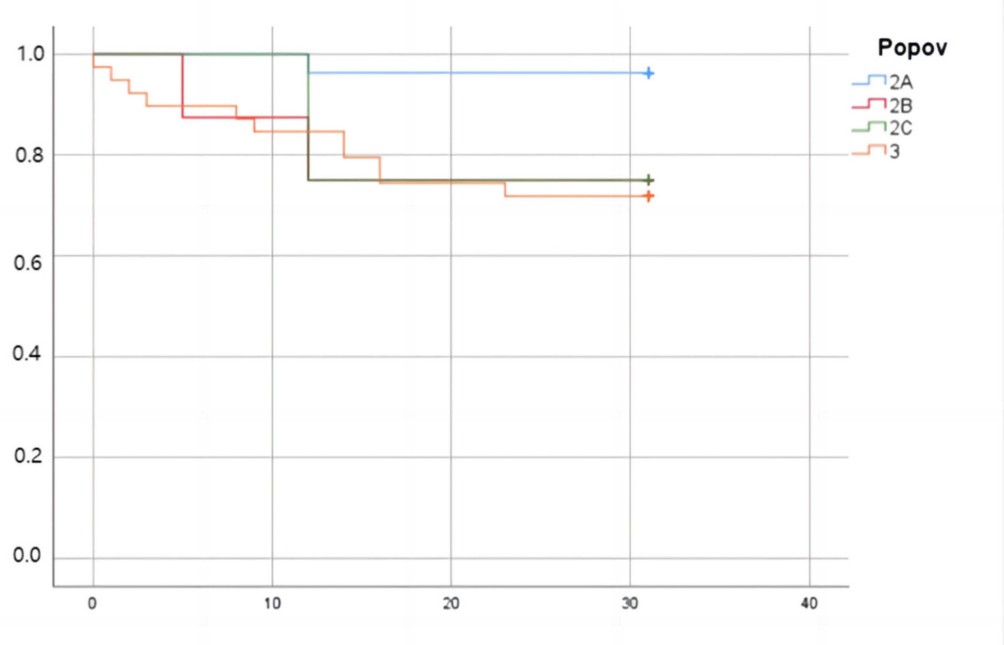

**Figure 6.** *Cont.*

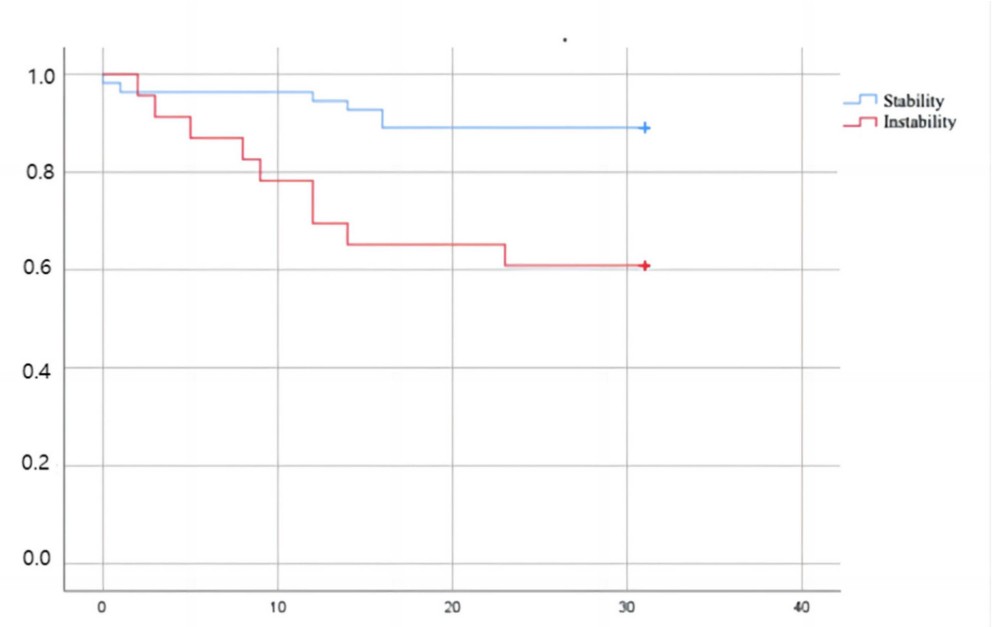

**Figure 6.** Kaplan-Meier curves for Popov categories and for hemodynamic status.

## 4. Discussion

SSTH incidence is rising annually because of the increasing number of patients undergoing ACT [1–5]. Surgery is the most commonly used method for the management of active bleeding. However, in recent years, TAE has demonstrated its central role in bleeding control with an increasing number of published case reports and case series. Furthermore, the surgical approach primarily aims to remove the large hematomas, in some cases removing the tamponade effect and increasing the risk of rebleeding [14–19].

Our study reports an extensive series of TAE procedures performed on patients with SSTH and active bleeding with anticoagulation impairment to evaluate the procedure's safety and efficacy and identify risk factors for death.

The distribution of hematoma locations, predominantly in the rectus sheath (53%), highlights the importance of considering this anatomical area when evaluating patients with coagulation impairment and clinically suspected bleeding. Anticoagulant therapy was the most prevalent cause of coagulation impairment in our population, with only 5 patients not receiving anticoagulant therapy, emphasizing the need for careful management of anticoagulation therapy in high-risk individuals. However, clinical management must always be tailored to the patient's circumstances. Considering the severity of the bleeding and the urgent nature of the situation, it was decided not to administer any antagonization treatment and to proceed with the interventional approach as expeditiously as possible.

The results of this study showed that TAE was technically feasible, safe, and effective. Immediate procedural success was achieved in all but one patient (98.7%), who died of bleeding-related causes several hours after the procedure, indicating the effectiveness of TAE in controlling the active bleeding. The most common TAE-related complication was acute kidney injury (AKI), which occurred in 31% of patients. However, the exact percentage of AKI due to contrast media toxicity is difficult to evaluate, especially considering that AKI can be a consequence of the bleeding itself or can derive directly from the patient's comorbidities. No direct correlation between contrast dose and renal damage has been established in this study. Kidney injury due to multifactorial causes probably accounts for a higher incidence of AKI in comparison with the findings obtained by other authors (10%) [21,22]. Only 2 patients (2.5%) were discharged with a new diagnosis of chronic kidney disease (CKD). Apart from AKI, the complication rate was low (3%) without long-term consequences, emphasizing TAE safety. Puncture site complications occurred only twice,

requiring stenting in one case. No complications related to embolization, such as ischemia or infection in the embolized territories or nontarget embolization, were observed. This result could be explained by using coils and resorbable materials, allowing collateral vessel development and eventually revascularization. These results align with previous findings obtained by other authors [19].

Embolization stopped the acute bleeding in most cases, as indicated by the low number of patients requiring a second TAE (16%). The high success rate of second transcatheter arterial embolization (TAE) procedures further supports the utility of this intervention in cases where initial embolization is insufficient.

The discrepancy between active bleeding observed in CTA and DSA suggests that CTA may not always accurately reflect the bleeding status at the time of angiography. This discrepancy may have implications for treatment decision-making and underscores the importance of timely and accurate diagnostic procedures.

Mortality for all causes was 19% at 30 days, which is lower but comparable to the literature (23%) [7]. Only 8% of patients died from causes directly related to bleeding, and death was rarely due to intractable hemodynamic shock. Instead, it was due to severe general conditions in a fragile patient [15], particularly low eGFR or hemodynamic instability at diagnosis, which were significant prognostic factors for our patients. The different TAE techniques did not have any different impact on the prognosis of the patients in the study, as the difference between those treated with reabsorbable embolic agents alone and with reabsorbable embolic agents combined with coils was not statistically significant.

According to the classification by Popov et al., we stratified our patients into four categories of risk. We chose, differently from previous studies in the literature, to perform TAE instead of conservative treatment on all patients with active bleeding, including type 2A patients. Although our results align with the algorithm proposed by Popov et al., we demonstrated that the mortality rate was comparable in patients belonging to types 2B, 2C, and 3 and significantly lower in patients belonging to the type 2A group (OR 9.2, 95% CI 1.1–74.5, $p < 0.05$). The 2A group was compared to other groups to assess whether ACT suspension had an effect on the self-limiting nature of bleeding, finding that there were no statistically significant differences in the self-limiting nature of bleeding based on drug suspension. This finding supports and evidences that even patients in group 2A required embolization treatment independently of the possibility of stopping ACT. Moreover, it underlines the necessity to intervene in these patients before the bleeding becomes complicated and spreads, thereby impacting the patient's clinical outcome. Moreover, no significant difference in the incidence of self-limited bleeding at DSA for Popov categories 2A and 2B-2C was evident, strengthening the fact that TAE is a safe and feasible treatment option also in 2A patients.

Complete fascial rupture, contraindications to ACT cessation, and the presence of intrinsic coagulation impairment were all reasonable risk factors for bleeding progression with a conservative approach [17]. Our study pointed out that they are essential prognostic factors even when TAE has been performed. In fact, as already stated, categories 2B, 2C, and 3, independently from TAE effectiveness, had a higher mortality rate, mainly due to previous comorbidities rather than TAE-induced complications.

The study has some limitations, including its retrospective design and single-center experience. The small sample size and lack of a control group also limit the generalizability of the findings. Additionally, the study did not evaluate long-term outcomes beyond the 30-day follow-up period.

Nonetheless, we demonstrated that TAE is both effective and safe in type 2A patients, and thus, it should be promptly considered for all patients with active bleeding seen on CT scans. Moreover, given that most type 2A patients were still bleeding during the DSA examination, which was performed with a median delay of 148 min after CTA, we propose that treating type 2A patients with TAE could potentially reduce blood transfusions and shorten hospital stays.

In conclusion, this study provides valuable insights into whether the use of TAE in patients with SSTH outweighs its potential complications, even for type 2A patients. It remains unclear whether type 2A patients could benefit from conservative treatment rather than TAE. Further research and larger studies are needed to better understand the optimal management strategies for different types of SSTH and to evaluate long-term outcomes.

**Author Contributions:** Conceptualization, D.F., L.P.M. and R.C.; methodology, R.F.G., E.F. and R.C.; software, D.L. and D.S.; validation, D.F. and E.F.; formal analysis, D.S.; investigation, D.F., E.F. and M.P.; resources, D.L. and M.P.; data curation, D.S. and D.L.; writing—original draft preparation, D.L. and M.P.; writing—review and editing, M.P. and E.F.; visualization, S.D.P. and R.F.G.; supervision, D.F. and E.F.; project administration, D.F. and E.F.; funding acquisition, E.F. All authors have read and agreed to the published version of the manuscript.

**Funding:** This study was not supported by any funding.

**Institutional Review Board Statement:** The study was conducted in accordance with the Declaration of Helsinki.

**Informed Consent Statement:** For this type of study, formal consent is not required. Consent for publication was obtained for every individual's data included in the study.

**Data Availability Statement:** The datasets used and/or analyzed during the current study are available from the corresponding author upon reasonable request.

**Conflicts of Interest:** The authors declare no conflict of interest. No benefits in any form have been received or will be received from a commercial party related directly or indirectly to the subject of this article.

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
