# Peer review of "Spontaneous Soft Tissue Hematomas in Patients with Coagulation Impairment: Safety and Efficacy of Transarterial Embolization"

_tomography, doi:10.3390/tomography9030089_

Round 1

Reviewer 1 Report

Dear Authors,

I’ve read with interest the paper entitled: “Spontaneous soft tissue hematomas and anticoagulant therapy: safety and efficacy of transarterial embolization”.

The authors accurately presented a case series of active-bleeding spontaneous soft tissue hematomas which had been embolized with a percutaneous transarterial approach. Even if no guidelines are available at the moment and further studies are needed, interesting interventional approach with good technical results has been presented also on Popov’s 2A patients (hemodynamically unstable with the possibility to stop anticoagulant therapy).

Manuscript is well written, following correct scientific methodology; however, a few points might be revised to make this paper clearer for the reader.

Major strengths

  1. interesting point on interventional approach on Popov’s 2A patient

Major weaknesses

  1. retrospective study
  2. relatively low number of patients

Title

-          As not all patients with SSTH were undergoing anticoagulant therapy (fig.4, Popov category 2C), I suggest a slightly modification of the title to make it more generic, such as “Spontaneous Soft Tissue Hematomas in patient with Coagulation Impairment: Safety and Efficacy of Transarterial Embolization”

Abstract

-          Generally ok. An explicit sentence about which were primary and secondary outcomes may add value in understanding the paper

Manuscript

-          Introduction l.43: give some examples of underlying diseases which can lead to SSTH

-          l. 46: as above, explicit “specific” comorbidities

-          interesting point: no significant difference in the incidence of self-limited bleeding at DSA for Popov categories 2A and 2B-2C; this point may add value to your interventional approach on 2A patients.

-          discussion l.257 “combined use of reabsorbable agents and coils was reserved for patients with confirmed active bleeding and super-selective catheterization of a target vessel”: so every patient with confirmed bleeding was treated by both resorbable agent AND coils? Why?

-          Discussion l.277 “higher…” some words are missing in this sentence.

Figures

-          if available, CTA of the patient in fig.2 may add value to the presentation of her diagnostic-therapeutical path

Author Response

Response to Reviewer 1 Comments

Dear Reviewer,

Thank you for giving us the opportunity to submit a revised draft of our manuscript. We appreciate the time and effort that you and the other reviewer have dedicated to providing your valuable feedback on our manuscript. We are grateful for your insightful comments on our paper. We have been able to incorporate changes to reflect most of the suggestions provided. We have highlighted the changes within the manuscript. Here is a point-by-point response to reviewers’ comments and concerns.

Point 1: Title, As not all patients with SSTH were undergoing anticoagulant therapy (fig.4, Popov category 2C), I suggest a slightly modification of the title to make it more generic, such as “Spontaneous Soft Tissue Hematomas in patient with Coagulation Impairment: Safety and Efficacy of Transarterial Embolization”

Response 1: Thank you for pointing this out. We agree with this comment and we proceeded in changing the title in “Spontaneous Soft Tissue Hematomas in patient with Coagulation Impairment: Safety and Efficacy of Transarterial Embolization” (see attached file, page 1, lines 2-3)

Point 2: Abstract Generally ok. An explicit sentence about which were primary and secondary outcomes may add value in understanding the paper

Response 2: Done. We have, accordingly, revised the abstract to emphasize this point adding this sentence: “Patient’s 30 day survival after TAE was considered as primary outcome; immediate technical success, the need for additional TAE and TAE-related complications were considered as secondary outcomes.” (see attached file, page 1, lines 24-26).

Point 3: Manuscript, Introduction l.43: give some examples of underlying diseases which can lead to SSTH

Response 3: We agree with this comment and we have incorporated your suggestion throughout the manuscript (please see the attached file, page 2, lines 53-54) in which we added “(e.g. infectious disease, chronic liver diseases, blood clotting disorders or cancer)”.

Point 4: l. 46: as above, explicit “specific” comorbidities

Response 4: As in response 3, we agree with this comment and we included in our manuscript specific comorbidities “(e.g. inflammatory conditions, aging or radiation therapy)” (see attached file, page 2, lines 56-57).

Point 5: interesting point: no significant difference in the incidence of self-limited bleeding at DSA for Popov categories 2A and 2B-2C; this point may add value to your interventional approach on 2A patients.

Response 5: Thank you for bringing this to our attention. We concur with your observation and have included this statement in our discussion, as indicated in the attached file (page 9, lines 297-298). This further strengthens our assertion that the interventional approach is a safe and feasible treatment option for 2A patients.

Point 6: discussion l.257 “combined use of reabsorbable agents and coils was reserved for patients with confirmed active bleeding and super-selective catheterization of a target vessel”: so every patient with confirmed bleeding was treated by both resorbable agent AND coils? Why?

Response 6: Thank you for pointing this out. We modified this sentence in “The combined use of reabsorbable agents and coils was considered as an option for patients with confirmed active bleeding when super-selective catheterization of a target vessel was obtained” to emphasize that the combined use of reabsorbable agent and coils was considered not in every patient, but only when a super-selective catheterization of a target vessel was achieved (see attached file).

Point 7: Discussion l.277 “higher…” some words are missing in this sentence.

Response 7: Done. We revised the sentence adding the missing words “mortality rate” (see attached file).

Point 8: Figures if available, CTA of the patient in fig.2 may add value to the presentation of her diagnostic-therapeutical path

Response 8: Regarding the CTA of the patient in Figure 2, we added the corresponding image, changing the figure caption (see attached file).

In addition to the above comments, all spelling and grammatical errors pointed out have been corrected and we proceeded in adding to embolic agents glue, which was missing.

Sincerely,

Dr. Eliodoro Faiella

Reviewer 2 Report

Thank you for the opportunity to review the paper:  "Spontaneous Soft Tissue Hematomas And Anticoagulant Therapy: Safety and Efficacy of Transarterial Embolization"

The authors report about a reasonable large cohort of SSTH -patients with  a history of anti-coagulation, platelet-inhibitors or other negative influences on the coagulation system. All patients received one or two transarterial embolizations. The method was safe and feasable in all subgroups.

Overall,  there is a need to thoroughly revise the results and discussion. The aspect of anticoagulation, which characterizes the title, is not consistently considered in the manuscript itself. Damage to kidney function is also only dealt with very superficially.

So, at least, data on the exact medication, antagonization (if applicable) and dose of contrast agents must be supplemented.

Lastly, Popov's classification is discussed several times. For the questions dealt with here, however, this does not bring clarity, but rather vagueness, because it does not reflect ACT in the last group.

Furthermore results and discussion have to be seperated more precisely.

Comments in details:

Title: "Spontaneous Soft Tissue Hematomas And Anticoagulant Therapy" - is misleading because 6 patients of 78 did not receive anticoagulants at all. Please rename the title more precisely and correct the first sentence of the abstract accordingly.

For the whole manuscript: Please decide to which level numbers will be written out or left as digits. 

Be consistent in the use of the term "Angiography". It would be more stringent, after an initial definition of CTA and DSA, to use only these abbreviations to avoid confusion

Misspelling in line 134    "catheter" instead of "catherer"

line 135 "... was confirmed on CTA..."  shouldn`t that mean "...was confirmed in angiography..." ?

line 142: What is meant by "...empiric temporary embolization..." ? Please specify.

Overall, there is no common thread in the analysis of the results and the discussion. The following questions should be answered:

1. What ACT was taken?

2. Were these antagonized and, if so, how quickly (even before the DSA)?

3. Discontinuation of anticoagulation alone won`t show any effect within the short time to DSA for most drugs (with the exception of intravenous heparinization). So why should there be a difference between group 2a and 2b in regard to self-limitation of the bleeding ?

4. What is the idea of comparing group 2a with groups 2b, 2c and 3?  In Popovs classification, there is no distinction in group 3 whether ACT is taken or not, so the aim for the research question of this paper is unclear (alternatively - think about changing the title!).

line 244 ff: When discussing contrast-induced kidney damage, it seems urgently necessary to show the relationship with the respective dose.

line 255-258 : "No statistically significant difference between embolizing agents was found. Reabsorbable embolic agents alone were used empirically when angiography did not confirm active bleeding. The combined use of reabsorbable agents and coils was reserved for patients with confirmed active bleeding and super-selective catheterization of a target vessel." This is a mixture of methods and results but not part of the discussion. 

line 265 f and 269 f: Again, these are results and should only be discussed in that chapter, but not any more mentioned in detail 

line 260: 16% necessity of a second DSA does not seem to be very small and is worth taking a look at the correlation between embolization material and success rate

Line 276: there is a word missing: "As a result, we demonstrated that type 2B, type 2C and type 3 patients had a higher  (?) than type 2A patients"

Legend Figure 6: "hemodynamic" instead of "hemodinamic"

Author Response

Response to Reviewer 2 Comments

Dear Reviewer,

Thank you for giving us the opportunity to submit a revised draft of our manuscript. We appreciate the time and effort that you and the other reviewer have dedicated to providing your valuable feedback on our manuscript. We are grateful for your insightful comments on our paper. We have been able to incorporate changes to reflect most of the suggestions provided. We have highlighted the changes within the manuscript. Here is a point-by-point response to reviewers’ comments and concerns.

Point 1: Title: "Spontaneous Soft Tissue Hematomas And Anticoagulant Therapy" - is misleading because 6 patients of 78 did not receive anticoagulants at all. Please rename the title more precisely and correct the first sentence of the abstract accordingly.

Response 1: Thank you for pointing this out. We agree with this comment and we proceeded, as suggested from reviewer 1, in changing the title in “Spontaneous Soft Tissue Hematomas in patient with Coagulation Impairment: Safety and Efficacy of Transarterial Embolization” and the first sentence of the abstract in “…with spontaneous soft tissue hematomas (SSTH) and active bleeding with anticoagulation impairment” (see attached file, page 1, lines 2-3, 21-22).

Point 2: For the whole manuscript: Please decide to which level numbers will be written out or left as digits.

Response 2: We agree with this comment and have incorporated your suggestion throughout the manuscript preferring to change letters to digits (see attached file, pages 1, 7, 9, lines 25-28, 224, 229, 238, 299).

Point 3: Be consistent in the use of the term "Angiography". It would be more stringent, after an initial definition of CTA and DSA, to use only these abbreviations to avoid confusion

Response 3: Thank you for pointing this out. We agree with you and proceeded in modifying the manuscript specifying if CTA or DSA was done as follows “If no active bleeding occurred during CTA, empiric temporary embolization was per-formed based on CTA imaging suspicion without prior confirmation of the exact origin location of the bleeding. Manual compression or a vascular closure device (Angio-Seal™; Terumo, Tokyo, Japan) achieved hemostasis at the puncture site. Data regarding the time of the first DSA run, the presence of active bleeding at CTA, the type of embolic agent used, and target vessels were recorded” (see attached file, pages 4-5, lines 128 – 129, 157-161).

Point 4: Misspelling in line 134    "catheter" instead of "catherer"

Response 4: Done. We revised the sentence correcting the misspelling (see attached file).

Point 5: line 135 "... was confirmed on CTA..."  shouldn`t that mean "...was confirmed in angiography..." ?

Response 5: Done. We revised the sentence correcting with the right proposition “if active bleeding was confirmed in CTA” (see attached file).

Point 6: line 142: What is meant by "...empiric temporary embolization..." ? Please specify.

Response 6: Agree. We have, accordingly, modified that sentence as follows “If no active bleeding occurred during CTA, empiric temporary embolization was performed based on CTA imaging suspicion without prior confirmation of the exact location of the bleeding. “ in order to clarify the concept of temporary embolization (see attached file).

Point 7: Overall, there is no common thread in the analysis of the results and the discussion. The following questions should be answered:

  1. What ACT was taken?
  2. Were these antagonized and, if so, how quickly (even before the DSA)?
  3. Discontinuation of anticoagulation alone won`t show any effect within the short time to DSA for most drugs (with the exception of intravenous heparinization). So why should there be a difference between group 2a and 2b in regard to self-limitation of the bleeding ?
  4. What is the idea of comparing group 2a with groups 2b, 2c and 3? In Popovs classification, there is no distinction in group 3 whether ACT is taken or not, so the aim for the research question of this paper is unclear (alternatively - think about changing the title!).

Response 7: Thank you very much for your evaluation regarding this topic. Regarding the anticoagulant therapy of the patients, out of the 62 patients receiving anticoagulant therapy, 37 were on heparin, 19 were on apixaban, and 6 were on warfarin. Only one patient was on antiplatelet therapy (cardioaspirin), and 10 patients were on both anticoagulant and antiplatelet therapy (heparin and cardioaspirin) (added to the attached file). As for point two, since the bleeding events were minor ones in an emergency setting, no antagonization was performed. Regarding point 3, we did not expect a difference between group 2A and group 2B in terms of self-limiting bleeding, but group 2B, being a group that cannot suspend therapy, has a worse prognosis and outcome. Moreover, group 2B has shown higher mortality precisely because, unable to stop the therapy, it experienced greater comorbidities. Lastly, regarding point 4, the reason for comparing group 2A with other groups was to assess whether therapy suspension had an effect on the self-limiting nature of bleeding. In reality, there are no statistically significant differences in the self-limiting nature of bleeding. This implies that even patients in group 2A require embolization treatment. In fact, it is necessary to intervene on these patients before the bleeding becomes complicated and spreads, thereby impacting the patient's clinical outcome.

Point 8: line 244 ff: When discussing contrast-induced kidney damage, it seems urgently necessary to show the relationship with the respective dose.

Response 8: Thank you for bringing this to our attention. We have thoroughly analyzed our dataset regarding the administered intraprocedural contrast, and we have observed a mean contrast amount of 140 ± 40 (mL) for all patients 50% diluited with saline solution (280 ± 80 mL of injected solution), with not statistically significant difference between the groups. However, as already mentioned in the article, it is important to note the reason why no specific value was reported. Correlating post-procedural renal damage solely with the intra-procedural administration of the contrast medium is challenging. These patients should be considered in an urgent setting, taking into account numerous comorbidities, which vary for each individual patient. However, after your report, we proceeded adding this data to the text (see attached file). If further modifications are required, we will add them as soon as possible.

Point 9: line 255-258 : "No statistically significant difference between embolizing agents was found. Reabsorbable embolic agents alone were used empirically when angiography did not confirm active bleeding. The combined use of reabsorbable agents and coils was reserved for patients with confirmed active bleeding and super-selective catheterization of a target vessel." This is a mixture of methods and results but not part of the discussion.

Response 9: Thank you for bringing this to our attention. We concur with your observation and we moved this sentences from the discussion respectively to results (“No statistically significant difference between embolizing agents was found”) and methods (“Reabsorbable embolic agents alone were used empirically when angiography did not confirm active bleeding. The combined use of reabsorbable agents and coils was reserved for patients with confirmed active bleeding and super-selective catheterization of a target vessel”) (see attached file).

Point 10: line 265 f and 269 f: Again, these are results and should only be discussed in that chapter, but not any more mentioned in detail

Response 10: Thank you very much for your evaluation regarding these two sentences. Regarding the first one, we have proceeded to move it to the appropriate section while still referencing it within the discussion, as it is an integral part of the discussion. Regarding the second sentence, in our opinion, it does present results but is also an integral part of the discussion. Therefore, we prefer it to remain as it is since it is discussed within this section. However, if it is not clear, we are prepared to modify/move that sentence.

Point 11: line 260: 16% necessity of a second DSA does not seem to be very small and is worth taking a look at the correlation between embolization material and success rate

Response 11: Thank you for this suggestion. We would like to highlight that the immediate technical success of the embolization was 98.7% . The 16% percentage includes patients with a second bleeding or with a bleeding not evaluated during the first angiography. To emphasize the high technical success rate, it is worth noting that only 8% of patients died during the follow-up period.

Point 12: Line 276: there is a word missing: "As a result, we demonstrated that type 2B, type 2C and type 3 patients had a higher  (?) than type 2A patients"

Response 12: Done. We revised the sentence adding the missing words: “mortality rate” (see attached file).

Point 13: Comments on the Quality of English Language Legend Figure 6: "hemodynamic" instead of "hemodinamic"

Response 13: Done. We revised the sentence correcting with the right proposition (see attached file).

In addition to the above comments, all spelling and grammatical errors pointed out have been corrected and we proceeded in adding to embolic agents glue, which was missing.

We look forward to hearing from you in due time regarding our submission and to respond to any

further questions and comments you may have.

Sincerely,

Dr. Eliodoro Faiella

Round 2

Reviewer 2 Report

The authors did improve the quality of the paper a lot, but still there are some sentences, which remain unclear. I refer to line-numbers in the pdf-file:

There were 78 patients included, all with a confirmed active bleeding in the CTA (line 107-109 and line 127f). Nevertheless the authors describe an action "If no active bleeding occurred during...CTA"  (line 147-152), please clarify! 

Again, in line 137-139: "If active bleeding was confirmed in CTA, the microcatheter tip was advanced as close to the bleeding before embolization"... there shouldn`t be any other patients!

(Point 5 of my former comments did not refer to the proposition "on" or "in" but to angiography/DSA instead of CTA - in the new version line 139)

line 239 no statistically siginificant difference (instead of "not").

beside one oversight in line 239, language is fine

Author Response

Dear Reviewer,

We are pleased that the implemented changes have been appreciated, and we would like to express our gratitude for your prompt response as well as for the constructive feedback. Here is a point-by-point response to new reviewers’ comments and concerns. We have highlighted the changes within the manuscript.

Point 1: There were 78 patients included, all with a confirmed active bleeding in the CTA (line 107-109 and line 127f). Nevertheless the authors describe an action "If no active bleeding occurred during...CTA"  (line 147-152), please clarify!

Response 1: Thank you very much for your evaluation. In this sentence, there was an extra word: “no”. We were referring to the situation when active bleeding is evident during DSA, so the microcatheter is brought as selectively as possible close to the bleeding source. Consequently, the sentence in point 1 has been modified as follows: "If active extravasation of i.v. contrast was confirmed on DSA, the microcatheter tip was advanced as close to the bleeding before embolization" (see attached file).

Point 2: Again, in line 137-139: "If active bleeding was confirmed in CTA, the microcatheter tip was advanced as close to the bleeding before embolization"... there shouldn`t be any other patients!

Response 2: Thank you for pointing this out. We agree with you and proceeded in modifying the manuscript specifying “DSA” instead of “CTA”. We revised the sentence as follows: “As stated in “Response 1”, no modification is necessary in our opinion regarding this sentence: ”If no active bleeding occurred during DSA, empiric temporary embolization was per-formed based on DSA imaging suspicion without prior confirmation of the exact origin location of the bleeding.” (see attached file).

Point 3: (Point 5 of my former comments did not refer to the proposition "on" or "in" but to angiography/DSA instead of CTA - in the new version line 139)

 Response 3: Perfect, we apologize for the misunderstanding. We have made the necessary modifications to the sentence as follows “ was confirmed on DSA” (see attached file).

Point 4: line 239 no statistically significant difference (instead of "not").

Response 4: Done. We revised the sentence correcting the misspelling “no statistically significant” (see attached file).

Sincerely,

Dr. Eliodoro Faiella
